

# Proliferation of non-linear excitations
# in the piecewise-linear perceptron

Antonio Sclocchi[1*] and Pierfrancesco Urbani[2]

**1** Université Paris-Saclay, CNRS, LPTMS, 91405, Orsay, France
**2** Université Paris-Saclay, CNRS, CEA, Institut de physique théorique,
91191, Gif-sur-Yvette, France

⋆ antonio.sclocchi@universite-paris-saclay.fr

## Abstract

We investigate the properties of local minima of the energy landscape of a continuous non-convex optimization problem, the spherical perceptron with piecewise linear cost function and show that they are critical, marginally stable and displaying a set of pseudo-gaps, singularities and non-linear excitations whose properties appear to be in the same universality class of jammed packings of hard spheres. The piecewise linear perceptron problem appears as an evolution of the purely linear perceptron optimization problem that has been recently investigated in [1]. Its cost function contains two non-analytic points where the derivative has a jump. Correspondingly, in the non-convex/glassy phase, these two points give rise to four pseudogaps in the force distribution and this induces four power laws in the gap distribution as well. In addition one can define an extended notion of isostaticity and show that local minima appear again to be isostatic in this phase. We believe that our results generalize naturally to more complex cases with a proliferation of non-linear excitations as the number of non-analytic points in the cost function is increased.



# 1 Introduction

Marginal stability of hard sphere packings at jamming has been the subject of an intensive line of studies in the last twenty years [2, 3]. This stream of works has culminated in the exact solution of the statistical mechanics of dense glassy hard spheres in infinite spatial dimensions [4]. This has allowed a detailed description of the critical behavior observed at the jamming transition point. In particular, the critical pseudogaps in the distribution of contact forces between spheres as well as the divergence of the gap distribution for small gaps have been completely characterized in infinite dimensions. Remarkably, the mean field predictions have been shown, within numerical precision, to hold down to two dimensional hard sphere packings, see [5] for a review, something that has pushed towards a statement about the upper critical dimension for the jamming transition to be two [6–8]. Furthermore, these predictions have been shown to agree with the real space scaling argument description of marginal stability of jammed packings [3].

The critical behavior observed at jamming was believed to be peculiar of the transition point. Instead, very recently it has been shown that there is nothing special about jamming. In [1,9] it was performed a systematic investigation of the properties of soft spheres interacting with a purely linear repulsive potential as well as a mean field version of the same optimization problem, namely the spherical perceptron with linear cost function. In particular it has been shown that in the jammed phase, when the potential energy is non-convex with respect to the degrees of freedom, both systems *self-organize* into marginally stable, critical configurations at finite energy density. The corresponding properties appear to be remarkably close to the ones of amorphous jammed packings of hard spheres implying that the criticality emerging at the jamming transition is not so special after all. In particular, local minima of the energy landscape are characterized by a set of non-linear excitations. These excitations correspond to the breaking of contacts between pairs of spheres, while the related relaxation mechanisms correspond to the formation of contacts. As a difference with respect to jamming, such excitations are richer in nature, because the system has more mechanisms to break or form a contact between two spheres. At variance with hard spheres at jamming, for jammed linear spheres one could have in addition to contacts becoming positive gaps also contacts becoming negative gaps. Conversely, the formation of new contacts may come from small overlaps or positive gaps. The abundance of these excitations depends on how many forces in the contact network have values close to the stability bounds. It is controlled by the behavior of the force density distribution near the bounds, which has power law behavior with universal critical exponents. Similarly, the formation of contacts is controlled by the abundance of small gaps between pairs of spheres, which has a power law behavior with corresponding critical exponents [3, 10].

Remarkably, the critical exponents controlling the excitations' density appear to be the same (within numerical precision) to the ones of the jamming point of hard spheres. It follows that jamming criticality is inherently linked to the non-analyticity of the interaction potential. In the jammed phase, this becomes evident since, despite the fact that the energy is positive, packings sit on minima in which there is an isostatic[1] number of spheres that just touch (contacts). Therefore jamming criticality, meaning the type of marginal stability found at the jamming transition, survives in the whole jammed glassy phase.

In this work we explore what happens if the interaction potential has several linear ramps with different slopes separated by non-analytic points. We show that if we consider a piecewise generalization of the linear potential studied in [1,9], we obtain again that the jammed phase of the corresponding optimization problem is made of marginally stable minima whose prop-

---

[1]A mechanical system can be in mechanical equilibrium if the number of constraining forces is greater or equal than the number of degrees of freedom. If they are equal, then the mechanical stability condition is marginally satisfied and the system is said to be isostatic.

erties are again very close to hard spheres at jamming. Remarkably, we get that isostaticity still holds but we need to extend its notion to include the fact that gaps can sit in different non-analytic points of the interaction potential. Furthermore we show that for each non-analytic point of the cost function, two pseudogaps emerge whose critical exponents appear to be the same as the ones controlling the jamming transition. This implies a *proliferation* of non-linear excitations that can trigger plastic events when the system is perturbed in some way [10]. Our results reinforce the fact that jamming criticality does not pertain only to the jamming point but it is rather related to two concomitant ingredients: the singular nature of the cost function and the non-convex nature of the problem.

## 2   The model

We consider the spherical perceptron optimization problem with a piecewise linear cost function. The model is a mean field model for the corresponding optimization problem for spheres interacting with piecewise linear cost function in finite dimensions. Despite the fact that the study we perform here can be extended verbatim to piecewise linear spheres, we leave this for future work. However, given the results of Ref. [9], we expect that the conclusions we will draw from the analysis of the perceptron problem will apply also to finite dimensional spheres.

The perceptron optimization problem [11, 12] is defined by an $N$ dimensional vector $\underline{x}$ which lives on the $N$-dimensional sphere $|\underline{x}|^2 = N$. In addition, one extracts $M = \alpha N$ $N$-dimensional random vectors $\underline{\xi}^\mu$ with $\mu = 1, \dots, M$. Every component of all these random vectors is a Gaussian random variable with zero mean and unit variance. Given the set of random vectors, also called patterns, and the state vector $\underline{x}$, one can define a set of gap variables defined as $h_\mu = \underline{\xi}^\mu \cdot \underline{x}/\sqrt{N} - \sigma$, being $\sigma$ and $\alpha$ control parameters of order one. The optimization problem is defined in terms of such gap variables. One constructs the cost function

$$H[\underline{x}] = \sum_{\mu=1}^{\alpha N} v(h_\mu), \tag{1}$$

and asks to find the value of $\underline{x}$ that minimizes it. In this work we consider the piecewise linear cost function defined as

$$v(h) = \begin{cases} -2h - H_0 & h < -H_0 \\ -h & h \in [-H_0, 0] \\ 0 & h > 0 \end{cases}, \tag{2}$$

where $H_0$ is a positive constant of order one that is taken to be fixed. In Fig. 1 we sketch the form of the corresponding potential. The model admits a satisfiable phase that happens when, given $\alpha$, one chooses a sufficiently small $\sigma$. In this case one can find a configuration of $\underline{x}$ such that $h_\mu > 0$ for all $\mu = 1, \dots, M$. Conversely, as soon as one increases $\sigma$, fixing $\alpha$, one finds a point (that may be algorithm-dependent) beyond which finding configurations where all gaps are positive becomes algorithmically impossible. This corresponds to the jamming transition of the model. It is clear that the properties of the configurations at jamming do not depend on the cost function, since up to jamming no negative gap is present. For this reason we are not interested in studying jamming which has been fully analyzed in [12, 13]. Instead we want to look at the system beyond the jamming point. In this case local minimization algorithms such as gradient descent get stuck in local or global minima, depending on the convexity of the problem. We want to characterize the properties of such minima. We note that the spherical perceptron problem with purely linear potential studied in [1] can be obtained from Eq. (2) by taking the limit $H_0 \to \infty$.

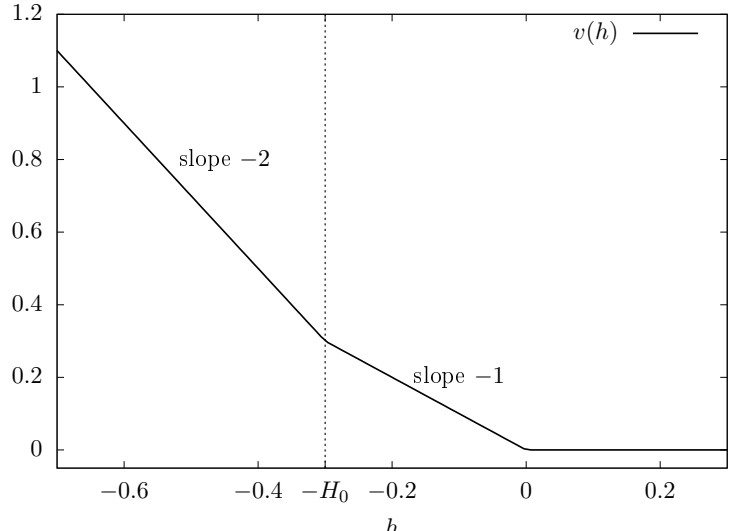

Figure 1: The piecewise linear cost function $v(h)$ defined in Eq. (2) where we set $H_0 = 0.3$.

In order to characterize local minima of the energy landscape we look at the distribution of gap variables. In the purely linear perceptron case of [1] it was found that the jammed non-convex/glassy phase contains minima where the distribution of gap variables contains a Dirac delta peak at $h = 0$. The weight of the peak is equal to $N$ which is the number of degrees of freedom in the problem. This implies that local minima have an isostatic number of gaps that are strictly equal to zero. This is the version of isostaticity that emerges when the cost function is purely linear. The presence of this isostatic peak is accompanied by an isostatic set of contact forces that can be thought as Lagrange multipliers needed to enforce that the corresponding gaps vanish.

In the present case we will show that we get a similar phenomenology. It is clear that in the glassy jammed phase the piecewise linear cost function induces the appearance of two Dirac delta peaks in the distribution of gaps centered in $h = 0$ and $h = -H_0$. Correspondingly one will have two sets of contact forces. The main questions we are interested in are: is the system going to be isostatic? What is the version of isostaticity that applies to this case? What are the properties of the contact forces? And what is the behavior of the distribution of gap variables in the jammed glassy phase of the model?

For what follows it will be convenient to introduce some notation to saparate the different types of gaps and contacts. We define the gaps that are less than $-H_0$ by $\mathcal{O}_< \equiv \{\mu : h_\mu < -H_0\}$. Furthermore we define by $\mathcal{O}_=$ the set of gaps that are in the interval $(-H_0, 0)$, namely $\mathcal{O}_= \equiv \{\mu : h_\mu \in (-H_0, 0)\}$. Moreover we define the set of contacts in $h = -H_0$ as $\mathcal{C}_{H_0} \equiv \{\mu : h_\mu = -H_0\}$ and the set of contacts in $h = 0$ as $\mathcal{C}_0 \equiv \{\mu : h_\mu = 0\}$.

## 3 Numerical simulations

In order to understand the properties of local minima of the model, we perform numerical simulations. We use the algorithm presented in [10], adapted now to the broken linear potential case, and the gradient descent minimizations of a regularized version of the potential, as done in [1].

In order to take into account the gaps that may end up being either exactly in zero or in

$-H_0$, we define the Lagrangian

$$
\begin{aligned}
\mathcal{L} = &\sum_{\mu \in \mathcal{O}_< \cup \mathcal{O}_=} \nu(h_\mu) - \sum_{\mu \in \mathcal{C}_0} f_\mu h_\mu \\
&- \sum_{\mu \in \mathcal{C}_{H_0}} f_\mu (h_\mu + H_0) + \frac{\mu}{2} (|\underline{x}|^2 - N) - pN\sigma \,,
\end{aligned}
\tag{3}
$$

where we have added the contact forces $f_\mu$ that take into account the gaps that eventually fall in $h = 0$ or in $h = -H_0$. In addition we have introduced a Lagrange multiplier $\mu$ that is needed to enforce the spherical constraint on the vector $\underline{x}$. The last term is added to change the control parameter from $\sigma$ to the pressure $p$. Given the Lagrangian $\mathcal{L}$, a local minimum satisfies the variational equations with respect to both $\underline{x}$ as well as the contact forces and $\sigma$ (which is no more a control parameter in the problem, and it is fixed essentially by the pressure).

The constitutive equations for local minima are

$$
\begin{aligned}
\mu x_i &= 2 \sum_{\mu \in \mathcal{O}_<} \frac{\xi_i^\mu}{\sqrt{N}} + \sum_{\mu \in \mathcal{O}_=} \frac{\xi_i^\mu}{\sqrt{N}} + \sum_{\mu \in \mathcal{C}_0 \cup \mathcal{C}_{H_0}} \frac{f_\mu \xi_i^\mu}{\sqrt{N}} \\
p &= \frac{2}{N} \sum_{\mu \in \mathcal{O}_<} 1 + \frac{1}{N} \sum_{\mu \in \mathcal{O}_=} 1 + \frac{1}{N} \sum_{\mu \in \mathcal{C}_0 \cup \mathcal{C}_{H_0}} f_\mu \\
h_\mu &= 0 \qquad \forall \mu \in \mathcal{C}_0 \\
h_\mu &= -H_0 \qquad \forall \mu \in \mathcal{C}_{H_0} \\
|\underline{x}|^2 &= N \,.
\end{aligned}
\tag{4}
$$

It is clear from the two slopes of the linear parts of the interaction potential that a physical solution to the variational equations (4) requires that

$$
\begin{aligned}
f_\mu &\in (0,1) \quad \forall \mu \in \mathcal{C}_0 \\
f_\mu &\in (1,2) \quad \forall \mu \in \mathcal{C}_{H_0} \,.
\end{aligned}
\tag{5}
$$

If a solution has contact forces that are outside the corresponding stability intervals, such solutions identify an unstable configuration.

We observe that, as it happens for the purely linear case [10], the Lagrangian $\mathcal{L}$ is effectively linear in all variables except for the term proportional to $\mu$. Therefore the convexity of the Lagrangian we are minimizing is due to the spherical geometry of $\underline{x}$ and is self-determined by the sign of $\mu$. If $\mu < 0$ we are in the non-convex phase with multiple minima and a glassy landscape, while if $\mu > 0$ we are in a convex phase with just one minimum. We will make use of this fact when arguing for isostaticity, see Eq. (8).

We choose to work at fixed $\alpha$ and to explore the jammed phase. We note that the value we have chosen for $\alpha$ corresponds to the situation in which jamming happens in a non-convex marginally stable situation and therefore it is in the same universality class as hard spheres [12]. Conversely if we choose $\alpha \leq 2$, jamming appears to be in a convex regime and is not critical anymore. Since we are interested in the properties of the non-convex/glassy phase, we fix $\alpha = 4$, for which jamming is obtained at $\sigma_J \simeq -0.42$. We explore the energy minima of the jammed phase in two ways. The first method is fixing a positive pressure $p > 0$ and performing a gradient-descent minimization of the smoothed Lagrangian[2]

---

[2]In the L-BFGS minimizations, we substitute the Lagrangian term imposing the spherical constraint, i.e. $\frac{\mu}{2}(|\underline{x}|^2 - N)$, with a quartic potential $\frac{\eta}{4}(|\underline{x}|^2 - N)^2$, where $\eta$ is a parameter chosen to be "large enough" in the numerical simulations. We empirically set $\eta = 500$. The Lagrange multiplier $\mu$ is recovered by $\lim_{\eta \to \infty} \eta(|\underline{x}|^2 - N) = \mu$.

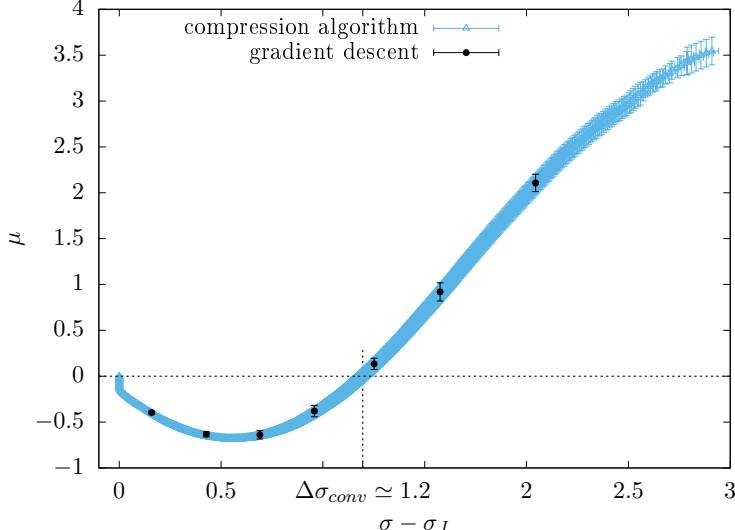

Figure 2: The Lagrange multiplier $\mu$ as a function of the distance to jamming. The jump observed at $\sigma = \sigma_J$ is due to finite size effects. Indeed in a finite system the configuration at jamming is stable for a finite amount of pressure before being destabilized and entering in the jammed phase [10]. The Lagrange multiplier is negative in the non-convex phase while it is positive when the landscape becomes convex. The data produced by the compression algorithm and the gradient-descent (L-BFGS) minimizations are consistent showing that despite the fact that they are different, they land on family of local minima that have very similar properties. The figure has been produced simulating the model, with both algorithms, at $\alpha = 4$ and $N = 512$, averaging over 100 samples. The errorbars represent sample to sample fluctuations.

$\mathcal{L}_\epsilon(\underline{x}, \sigma) = \sum_\mu v_\epsilon(h_\mu) + \frac{\mu}{2}(|\underline{x}|^2 - N) - pN\sigma$, where the $\epsilon$-regularized potential $v_\epsilon(h)$ corresponds to $v(h)$ with the singularities regularized by quadratic parts with curvature $1/\epsilon$:

$$v_\epsilon(h) = \begin{cases} -2h - H_0 & h < -H_0 - \frac{\epsilon}{2} \\ -h + \frac{1}{2\epsilon}(h + H_0 - \frac{\epsilon}{2})^2 & h \in [-H_0 - \frac{\epsilon}{2}, -H_0 + \frac{\epsilon}{2}] \\ -h & h \in [-H_0 + \frac{\epsilon}{2}, -\frac{\epsilon}{2}] \\ \frac{1}{2\epsilon}(h - \frac{\epsilon}{2})^2 & h \in [-\frac{\epsilon}{2}, \frac{\epsilon}{2}] \\ 0 & h > \frac{\epsilon}{2} \end{cases} . \tag{6}$$

It is evident that $\lim_{\epsilon \to 0} v_\epsilon(h) = v(h)$ and that the derivatives of the quadratic parts of $v_\epsilon(h)$ provide the contact forces through $\lim_{\epsilon \to 0}(h_\mu - \epsilon/2)/\epsilon = -f_\mu$ for $h_\mu \in [-\epsilon/2, \epsilon/2]$ and $-1 + \lim_{\epsilon \to 0}(h_\mu + H_0 - \epsilon/2)/\epsilon = -f_\mu$ for $h \in [-H_0 - \epsilon/2, -H_0 + \epsilon/2]$. We find the minima of $\mathcal{L}$ at fixed $p$ by minimizing $\mathcal{L}_\epsilon$ with the L-BFGS algorithm [14] and performing an annealing on the parameter $\epsilon$ to go to $\epsilon \to 0$. The gaps whose values end up in the $\epsilon$-windows around 0 and $-H_0$ form the sets $\mathcal{C}_0$ and $\mathcal{C}_{H_0}$ respectively. The second method consists in finding the jamming point[3] $\sigma_J$ and progressively compressing the system as in [10]. The minima explored by the two methods show statistical properties that are in perfect agreement.

In Fig.2 we plot the behavior of the Lagrange multiplier $\mu$ as a function of $\sigma - \sigma_J$ being $\sigma_J$ the jamming point. It is clear that as soon as we enter the jammed phase, the landscape

---

[3]We do it by performing an L-BFGS minimization of $\mathcal{L}_\epsilon$ as described in the text, but using a pressure $p$ small enough (see Ref. [10]).

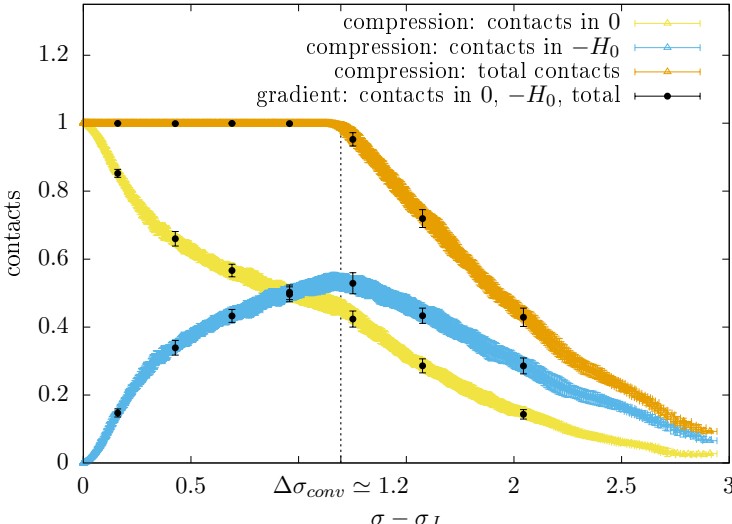

Figure 3: The number (normalized by $N$) of contacts in $h = 0$, meaning $c_0$, in $h = -H_0$, meaning $c_{H_0}$, and their sum. We clearly see that in the whole interval in which the system is in the glassy/non-convex phase, the total amount of gaps in the two non-analytic points of the cost functions is isostatic. Isostaticity is lost when the system is in the convex phase. Also in this case, as for the purely linear cost function, we notice that, in the glassy phase, the sample to sample fluctuations away from isostaticity are essentially absent. The data produced by the compression algorithm and the gradient-descent (L-BFGS) minimizations are consistent showing again that the minima explored by the two algorithms have similar properties. The figure has been produced simulating the model, with both algorithms, at $\alpha = 4$ and $N = 512$, averaging over 100 samples. The errorbars represent sample to sample fluctuations.

is strictly non-convex being the Lagrange multiplier negative. When compressing the system further, it undergoes a topology trivialization transition at $\sigma_{conv} - \sigma_J = \Delta\sigma_{conv}$ ($\simeq 1.196$ for $\alpha = 4$) towards a convex phase where the landscape is made of just one unique minimum and the Lagrange multiplier $\mu$ becomes positive. The behavior of $\mu$ is the same both for configurations explored through the compression algorithm and for those found by L-BFGS minimizations. We expect that this transition can also be found by analyzing the problem with the replica method and corresponds to the point where replica symmetry breaking appears (coming from the convex/non-glassy phase). This behavior mirrors the one found for the purely linear case [1, 10].

We now focus on the properties of the local minima in the glassy phase. We first measure the cardinality of the sets $\mathcal{C}_0$ and $\mathcal{C}_{H_0}$. This is plotted in Fig. 3 where we show the data for $|\mathcal{C}_{H_0}|/N = c_{H_0}$, $|\mathcal{C}_0|/N = c_0$ and the total number of contacts. At the beginning of the compression protocol the system contains $N$ gaps in zero and therefore the system is isostatic with a number $|\mathcal{C}_0| = c_0 N = N$. As soon as we enter the jammed phase, contacts in $-H_0$ start to appear. Remarkably we find that

$$|\mathcal{C}_0| + |\mathcal{C}_{H_0}| = N , \tag{7}$$

which implies that the system is isostatic only globally. The number of gaps in $h = 0$ or in $h = -H_0$ fluctuates but the total sum is equal to the degrees of freedom in the problem. Remarkably the sample to sample fluctuations of $c_0$ and $c_{H_0}$ seem to be normal yet completely anticorrelated in order to have Eq. (7) satisfied even at finite $N$. The system self-organizes in such a way that only the sum of the number of gaps in zero and $-H_0$ is isostatic.

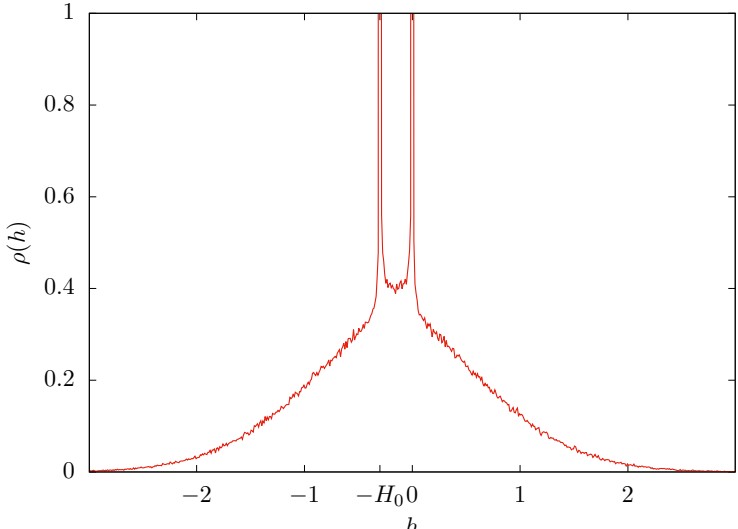

Figure 4: The empirical probability distribution function of the gap variables, obtained at pressure $p = 4$ and for $\alpha = 4$ and $N = 2048$.

To understand this fact we use the following argument. Let us imagine that we smooth out the non-analytic corners in the cost function of Eq. (2) by two small quadratic interpolation parts. Let us denote by $\epsilon$ the amplitude of the interpolated region. As for the purely linear case [1], the smoothing removes the degeneracy of the contacts and allows for a real space description of the contact forces that appear as the gap contained within the smoothed regions. Since now the cost function admits an harmonic expansion, we can define the corresponding (rescaled) Hessian, as done in the appendix B of Ref. [9]. Remarkably it takes contribution from both the contacts in $h = 0$ as well as the ones in $h = -H_0$ and it is given by

$$\epsilon \frac{\partial^2 \mathcal{L}_\epsilon}{\partial x_i \partial x_j} = \frac{1}{N} \sum_{\mu \in \mathcal{C}_0 \cup \mathcal{C}_{H_0}} \xi_i^\mu \xi_j^\mu + \epsilon \mu \delta_{ij}, \tag{8}$$

which is, neglecting correlations, a Wishart random matrix shifted on the diagonal [15]. In the glassy phase where $\mu < 0$ we need to have that the Wishart content of the Hessian matrix should be full-rank in order to have stable minima. Therefore we have that

$$|\mathcal{C}_0| + |\mathcal{C}_{H_0}| \geq N. \tag{9}$$

If marginal stability holds, the bound is saturated and we get isostaticity [3]. This argument tells that the number of contacts in $h = 0$ and $h = -H_0$ can fluctuate but in a correlated way in order to enforce Eq. (9).

Now we turn to the analysis of the force and gap distribution. We define the empirical distribution of gap variables as

$$\rho(h) = \frac{1}{M} \sum_{\mu=1}^{M} \delta(h - h_\mu). \tag{10}$$

In Fig.4 we plot the histogram of $\rho(h)$ at $p = 4$ which corresponds to the point where $c_0 \sim c_{H_0}$. It is clear from this qualitative picture that the two Dirac delta functions in $h = 0, -H_0$ are surrounded by four power law divergences.

In order to characterize those divergences, in Fig.5 we plot the cumulative distribution function of the gaps, starting from $h = 0$ and $h = -H_0$. Within our numerical precision we

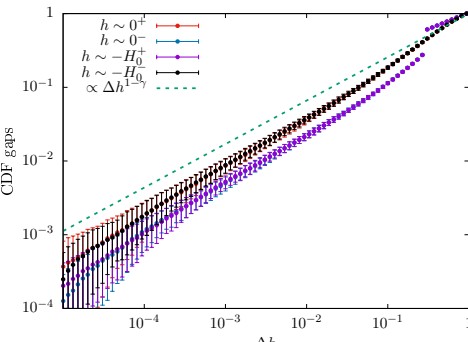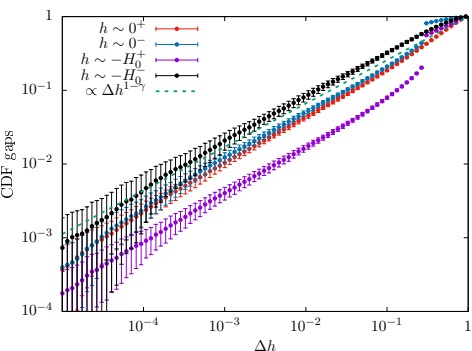

Figure 5: *Left panel:* The normalized cumulative distribution functions (CDF) of the gaps on both sides of both Dirac delta peaks at $h = 0$ and $h = -H_0$. We define the CDF as $\int_0^{\Delta h} \rho(h')dh'/\int_0^\infty \rho(h')dh'$ for $h \sim 0^+$, $\int_0^{-\Delta h} \rho(h')dh'/\int_0^{-\infty} \rho(h')dh'$ for $h \sim 0^-$, $\int_{-H_0}^{-H_0+\Delta h} \rho(h')dh'/\int_{-H_0}^\infty \rho(h')dh'$ for $h \sim -H_0^+$, $\int_{-H_0}^{-H_0-\Delta h} \rho(h')dh'/\int_{-H_0}^{-\infty} \rho(h')dh'$ for $h \sim -H_0^-$. $\Delta h$ represents $|h|$ for $h \sim 0^\pm$ and $|h + H_0|$ for $h \sim -H_0^\pm$. The fact that the curves almost coincide in pairs is not due to any physical reason, but simply to the fact that for the chosen point of the phase diagram the distribution of gaps is approximately symmetrical with respect to $h = -H_0/2$ (see Fig.4). The prefactors of the power laws are not in general equal and depend on the pressure $p$. On the right panel we show the same quantities for pressure $p = 2$ where the empirical symmetry is lost and prefactors appear to be different. The plot has been produced by L-BFGS minimizations at $p = 4$ for $\alpha = 4$ and $N = 2048$, averaged over 100 samples. The errorbars represent sample to sample fluctuations.

clearly see that

$$\rho(h) \sim \begin{cases} A_0^+ h^{-\gamma} & h \to 0^+ \\ A_0^- |h|^{-\gamma} & h \to 0^- \\ A_{H_0}^+ (h + H_0)^{-\gamma} & h \sim -H_0^+ \\ A_{H_0}^- |h + H_0|^{-\gamma} & h \sim -H_0^- \end{cases}, \tag{11}$$

where the exponent $\gamma \simeq 0.41\dots$ coincides (within our numerical precision) with the one characterizing the distribution of positive gaps at the jamming transition point [16, 17] and the $A$s are constants.

Finally we look at the contact forces. In Fig.6 we plot the empirical distribution of contact forces both in $h = 0$ and in $h = -H_0$. We clearly see that there are four pseudogaps appearing close to the edges of the support of $f_\mu$.

In order to quantitatively analyze the behavior close to the four edges of the stability supports, we look at the cumulative distribution functions that we plot in Fig.7. We clearly see within our numerical precision that around the edges of the stability supports the force distribution has four pseudogaps

$$\rho(f) \sim \begin{cases} B_0^+ f^\theta & f \to 0^+ \\ B_1^- (1-f)^\theta & f \to 1^- \\ B_1^+ (f-1)^\theta & f \to 1^+ \\ B_2^- (2-f)^\theta & f \to 2^- \end{cases}, \tag{12}$$

where the $B$s are constants of order one and the exponent $\theta = 0.42\dots$ is close to the one controlling the small forces at the jamming transition point [16, 17].

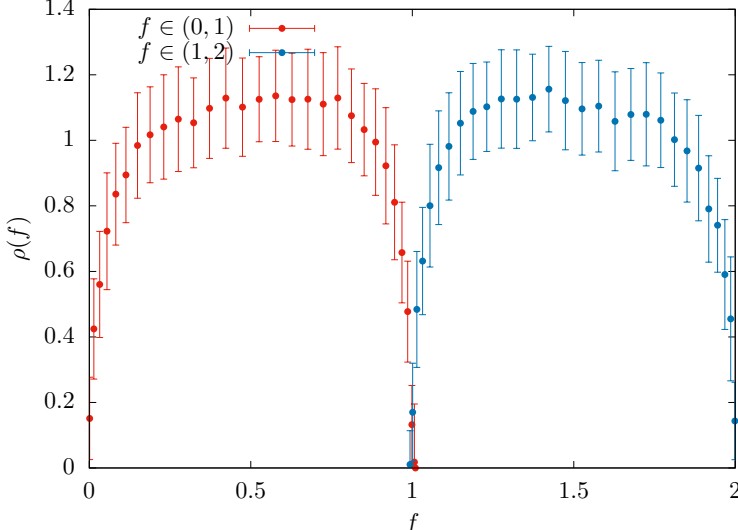

Figure 6: The empirical probability distribution function of the contact forces. In red we plot the ones corresponding to the gaps at $h = 0$ while in blue we plot the ones corresponding to the gaps at $h = -H_0$. The fact that the two pdfs appear rather similar is mainly due to the fact that we have measured such distribution at $p = 4$ where $c_0 \simeq c_{H_0}$. The data have been produced by L-BFGS minimizations at $p = 4$ for $\alpha = 4$ and $N = 2048$, averaged over 100 samples. Errorbars are obtained from sample to sample fluctuations.

## 4 Discussion

All in all, our numerical results suggest that again the glassy phase is isostatic and marginally stable. At variance with the purely linear potential and the jamming transition, here we have four pseudogaps characterizing contact forces and four power law divergences in the gap distribution. This implies a proliferation of non-linear excitations due to the fact that any perturbation can open and close all sorts of contacts. We believe that perturbing local minima of the system, as it happens for the purely linear potential case [10], will lead to system spanning avalanches and crackling noise [3]. This is a manifestation of the emergent self-organized criticality of the non-convex/glassy phase.

It is clear that our results may be generalized by adding more piecewise linear terms to the cost function. For each point where the potential has a kink, the corresponding gap distribution will get a Dirac delta peak. The argument about the stability of local minima suggests that in the non-convex phase, isostaticity is required to ensure marginal stability of local minima and that the isostatic condition involves a global sum-rule of the number of gaps that end up in one of the kinks of the cost function. This global topological constraint will enforce critical pseudogaps on both forces and gaps for each kink giving rise to a proliferation of non-linear excitations.

We expect, based on our experience with linear spheres [9], that the same results will hold for spheres interacting with piecewise linear potential down to two dimensions. In this case, one has the additional possibility to have localized non-linear excitations whose density decreases when increasing the packing fraction [9]. Moreover we expect that dense piecewise linear spheres, at finite temperature, will display strong Gardner phenomenology [18] upon cooling. Finally it would be interesting to see what happens for deeper models beyond the perceptron architecture [19–21] as well as more complex constraint satisfaction problems [22].

Beyond the isostaticity argument, an analytical understanding of the critical exponents

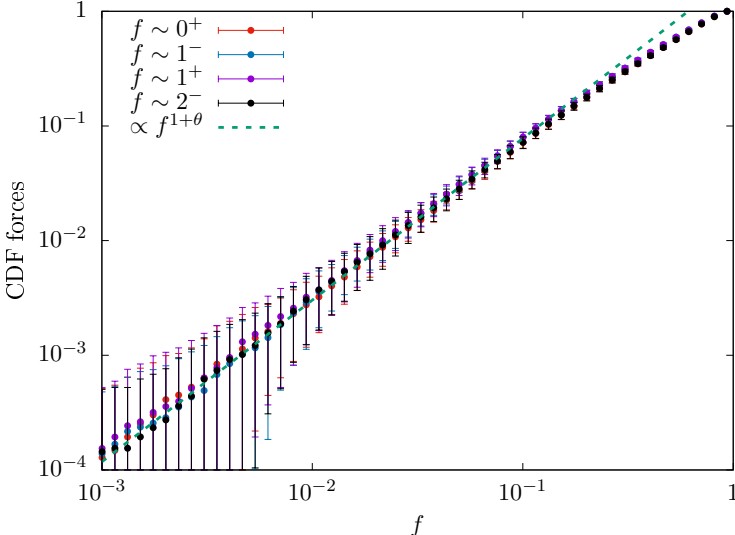

Figure 7: The cumulative distribution functions for the contact forces close to the edges of their stability supports. They are defined as $\int_0^f \rho(f')df'/\int_0^1 \rho(f')df'$ for $f \sim 0^+$, $\int_1^{1-f} \rho(f')df'/\int_1^0 \rho(f')df'$ for $f \sim 1^-$, $\int_1^{1+f} \rho(f')df'/\int_1^2 \rho(f')df'$ for $f \sim 1^+$, $\int_2^{2-f} \rho(f')df'/\int_2^1 \rho(f')df'$ for $f \sim 2^-$. We see that the apparent prefactors look very similar and the dots are rather one onto the other, because we measured the forces at the rather symmetric point where the number of gaps in zero and in $-H_0$ is roughly the same. We do not expect such prefactors to be universal but to depend on the point of the phase diagram where local minima are probed. The data have been produced by L-BFGS minimizations at $p = 4$ for $\alpha = 4$ and $N = 2048$, averaged over 100 samples. Errorbars are obtained from sample to sample fluctuations.

arising in each kink of the interaction potential requires the replica treatment of the model. Following [1], we would expect that as soon as the model has a ground state which has a continuous RSB solution at least close to the leaves of the ultrametric tree of pure states [23], a generalization of the scaling theory developed in [1] for the fullRSB equations should give rise to the exponents of the jamming transition, in agreement with the current numerical simulations. We leave the investigation of this aspect to future works. Despite the fact that however this replica approach holds strictly speaking for the ground state of the problem, as it happens for other problems, notably the Sherrington-Kirkpatrick model [23], it gives a prediction for the critical exponents arising in local minima that are obtained with greedy gradient descent algorithms as the ones we are using. While the derivation of such exponents from a purely dynamical/algorithmic perspective is an open problem, message passing algorithms are expected to be tracked by such replica scaling theory [24, 25], see also [26], and therefore to show the criticality we found here. This theoretical analysis points to the fact that the critical behavior is inherited from the non-analyticities of the cost function. Finally it would be interesting to understand what happens if one considers cost functions that have different types of non-analyticities and whether this could give rise to different critical behaviors. As an example one could look at potentials displaying infinite contact forces in a finite number of points (for example one could consider $v(h) = \sqrt{|h|}\theta(-h)$). We leave this problem for future work.

## Acknowledgments

We warmly thank Silvio Franz for very useful discussions. A.S. acknowledge a grant from the Simons Foundation (No. 454941, Silvio Franz). P.U. acknowledges the support of 'Investissements d'Avenir' LabEx PALM (ANR-10-LABX-0039-PALM).

**Funding information** This work was supported by "Investissements d'Avenir" LabExPALM (ANR-10-LABX-0039-PALM) and by the Simons foundation (grants No. 454941, S. Franz).

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
