# Peer review of "Proliferation of non-linear excitations in the piecewise-linear perceptron"

_SciPost Physics, doi:SciPost Phys. 10, 013 (2021)_

## Round 1 · Referee Report · Thibaud Maimbourg (Referee 1) · 2020-11-19

Strengths

1- The results deepen the understanding of jamming criticality in the overcompressed phase due to discontinuous forces. 2- The paper is clear and succinct. 3- The numerical results are convincing. 4- A plausible application to non-mean-field models (interacting spheres in 2D or 3D) is discussed.

Weaknesses

1- A weakness contained in the related strength. The paper builds on previous studies. Theoretical arguments are given as a transposition of the purely linear case without details and the numerical procedure is explained in previous works. As a consequence, the paper requires knowledge of the related works, which arguably is a right choice here, provided minor adjustments. 2- The figures' legibility and relation to the text could be improved.

Report

In this paper, the authors build on previous studies in the last years, by themselves and others, to investigate the consequences of non-analyticities in the cost function of the spherical perceptron optimization problem. Previous results have indeed shown that for a case of continuous forces $-v'(h)$, jamming criticality is somewhat fragile (due to null forces at contact) but its peculiar properties (related to isostaticity) get extended considering a linear potential (i.e. a finite sustaining force at zero gap). The paper studies the effects of additional such discontinuities of the forces (i.e. a piecewise linear potential at negative gap). In previous studies the non-analyticity of the potential was only located at zero gap, and contacts could be defined as constraints that are marginally unsatisfied. Here additional force discontinuities inside the region of unsatisfiability of the constraints lead to a definition of several "contact" locations (at each discontinuity), a notion of stability represented by the sum rule Eq.(8), corresponding to isostaticity when marginally satisfied, and leading to several delta peaks of the gap distribution surrounded by power laws, as well as four pseudogaps in the contact forces. The paper ends with a qualitative discussion of the corresponding emergent excitations, generalizations of the model, a plausible replica solution and application to interacting spheres.
The main tools are numerical simulations (using an algorithm developed in Ref.10 for a purely linear cost function at negative gaps) and theoretical arguments (a transposition of the pure linear case previously studied).

I found the paper well written, interesting and making a stronger point for non-convexity and non-analyticity of the cost function as ingredients for jamming phenomenology, that suggests a direction for future research. Considering the papers' strengths and weaknesses, minor modifications are required, mostly of informative character to the reader and to improve readability of the paper.

Requested changes

1- What is meant by convexity should be defined early on (convexity of the cost function with respect to certain variables). Similarly, isostaticity should be defined early in the text. The first appearance of minima in the text (properly defined in the abstract however) should be (re)defined as minima of the cost function / energy landscape.

2- As was correctly done for Fig. 4, the other figures should indicate the parameters of the model instead of only numerical numbers. Namely, in Fig.1, there should be a label $-H_0$ on the $h$ axis at -0.3 with a vertical dashed line indicating the discontinuity and labels indicating the slopes -1 or -2 of the ramps. This is to give a visual representation to Eq.(2) as was intended by the authors. In Fig. 2 a $\sigma_{\rm convex}$ or $\Delta\sigma_{\rm convex}=\sigma_{\rm convex}-\sigma_J$ (or any better name that the authors see fit) should be defined, its numerical value in the present case given and labelled on the horizontal axis. This could be used in the text when talking about the topology trivialization transition. In Fig. 3 this label should be used again on the horizontal axis, perhaps with a vertical dashed line to indicate the isostaticity breakdown. In Fig. 5 (in the legends there are minus signs missing $h\sim -H_0$) and the horizontal axis should be labelled as some $\Delta h$ which could be defined in the caption.

3- In the second paragraph (p.1), what is meant by excitations "richer in nature" ?

4- In the next paragraph below Eq.(5) about the sign of $\mu$, a connection to Eq.(7) should be made.

5- Information about the number of patterns numerically sampled must be included.

6- Concerning Eq. (7), and referring to Eq.(3) of Ref.1, for the reader I guess it would be clearer to substitute $\cal H_{ij}$ by $\varepsilon\frac{\partial^2\cal L_\varepsilon}{\partial x_i\partial x_j}$ where $\cal L_\varepsilon$ could be explained as the regularized version of Eq.(3) of the present paper. Appendix B of Ref. 9 could be cited.

7- Could the authors comment on the fact that the exponents $\gamma$ and $\theta$ on Eqs. (11) are the same on the four sides of the contacts? Besides, in Fig. 5, is there a physical reason for the blueand violet curves to almost coincide? Could it be a consequence of the fact that these gaps belong to $\cal O_=$? Does it mean the following relation $A_0^-=A_{H_0}^+$ for the prefactors? (the very small $\Delta h$ points may not support this but it is hard to tell due to the error bars). Same question for the red and black curves. Is the discontinuity in the violet curve related to starting sampling gaps around $\Delta h=h+H_0=H_0=0.3$ i.e. $h=0$, meaning gaps closer to the other contact in $h=0$? Similarly, Fig. 7 seems to indicate equal prefactors in Eq. (11), would there be a physical reason for this?

8- In the discussion, reference(s) should be added concerning the possibility of localized non-linear excitations in spheres, and related works should be cited in Ref.18 (if the authors agree on their relevance), such as M. Geiger, S. Spigler, S. d’Ascoli, L. Sagun, M. Baity-Jesi, G. Biroli and M. Wyart, Phys. Rev. E 100, 012115 (2019) H. Yoshino, SciPost Phys. Core 2, 005 (2020)

9- In the final sentence it is unclear what are the other types of non-analyticities referred to. It seems the present phenomenology is linked to discontinuities in the forces, i.e a discontinuity in the first-order derivative. All higher-order singularities seem irrelevant. Maybe the authors have in mind something like a square root singularity (infinite force in a point)?

10- Some typos should be fixed by careful reading. Examples are: In Eq. (4) the optimization on $f_\mu$ should display $h_\mu=-H_0$ in $\mathcal C_{H_0}$ as well. The first sentence in second column of p.2 can be simplified. The authors can search occurence of "contatcs", "oder", "incresing".

11- This comment is not properly speaking a requested change, although the answers could be included in the paper if the authors deem it relevant. Do the authors expect any influence of the inner discontinuity at $-H_0$ on the phase diagram of the model with respect to the phase diagram derived in the pure linear case? It seems not as far as I can tell in the text. Is there in this respect a role played by the unsatisfied constraints which are not "contacts", i.e. in $\cal O_{<,=}$ ? They are present in the optimization of Eq. (4).

---

## Round 1 · Referee Report · Anonymous (Referee 2) · 2020-12-6

Strengths

  1. The authors study an interesting variant of the spherical perceptron problem in a non-convex / glassy regime, as a model for jamming of hard spheres in finite/low dimension.

  2. They reinforce a previous discovery that several criticality properties associated with the jamming (SAT/UNSAT) transition are carried by the system in the entire jammed non-convex phase.

  3. They investigates the link between the number of singularity points of the cost function and the form of isostaticity of the model in the jammed phase, and identify power laws in the gap and force distribution in the vicinity of the singularities.

Weaknesses

  1. The study is not accompanied by a supporting theory. The authors claim that their results can be backed by replica computations, but they do not perform said computations. This being said, their claim is credible given earlier theoretical work on the linear perceptron in ref [1].

  2. In absence of convexity, the properties of the local minima found are in principle very algorithm-specific. A skeptic would claim that all the results in the paper are specific to the BFGS algorithm (the algorithm used by the authors), and are far from describing ground states...

Report

Summary:

The authors study an interesting variant of the spherical perceptron problem in a non-convex / glassy regime where the cost function to be minimized is piecewise linear with two singularities. Previous work (ref [1]) studied the 'linear' perceptron model where the cost function is max(-x,0), i.e., has one singularity at zero, and discovered that the system is (almost) isostatic in the entire RSB region of the jammed phase, in the sense that the number of marginally satisfied constraints (i.e., where the gaps are equal to zero) is N(1 - o_N(1)).

This paper considers a piecewise linear cost function with two singularities: one at 0 and one at -H_0. The authors make a similar empirical observation: there are about N gaps equal to either -H_0 or 0 in the RSB region of the jammed phase. An interpretation is that the system is isostatic only globally. (Note that the fluctuations in the above two groups must then be anticorrelated.)

Moreover, the distribution of gaps, in addition to having a Dirac delta at each singularity, is found to exhibit four power laws, one on each side of each singularities. Experiments show that the exponent governing these powers laws is the same, and is given by the power law found at the jamming transition! The authors also study the distribution of forces at the contact points (i.e., the Lagrange multipliers of the gaps) and find similar power laws.

Comments:

This papers is an interesting empirical study on the nature of jamming criticality. The authors use the perceptron model as a good mean-field proxy for hard spheres in low dimension (this model is also worth studying for its own sake, as a mathematical model of high-dimensional random geometry).

The paper makes several interesting findings, all of which deserve further study. The finding that the system distributes an isostatic number of gaps on the singularity points of the cost function seems to be new, and is quite intriguing.

Requested changes

  1. As a non expert in this area, I found the paper relatively easy to read and understand. However, I did not understand the claims about 'proliferation of non-linear excitations', 'avalanches and crackling noises' and their relationship to the power laws found in the gap distribution. Perhaps the authors can be more pedantic and expand further on this.

  2. The notion of isostaticity deserves to be defined more precisely. The claims in the paper seems to be only approximate: c_0 + c_{-H_0} = 1 - o_N(1), as opposed to the stronger claim that the number of marginally satisfied constraints is exactly equal to N (or N-1). The authors can perhaps clarify which form they are claiming.

  3. See also points raised in Weaknesses (derivation of critical exponents from replica theory, the algorithmic issue.)

---

## Round 2 · Author Response

Dear Editor,

we would like to resubmit our manuscript entitled: "Proliferation of non-linear excitations in the piecewise-linear perceptron" for publication in Scipost Physics.

We would like to warmly thank the reviewers for the critical assessment of the manuscript. We found all the comments and requested changes extremely useful and we feel that the manuscript has really improved after the revision. We would like to express our gratitude to both of them. In the following we will respond to all the points raised by the reviewers.

REPORT 1.

First of all we would like to warmly thank the reviewer for the det ailed and complete critical assessment of our work. All the comments and requested changes have allowed us to improve the quality of the manuscript as well as the clarity of our presentation. We are very grateful for that.

In the following we will respond to the points raised by the reviewer.

Requested changes:

1) In the introduction we have written "In particular it has been shown that in the jammed phase, when the potential energy is non-convex with respect to the degrees of freedom, both systems \emph{self-organize} into marginally stable, critical, configurations at finite energy density." and added the footnote 1 to explain the concept of isostaticity.

2) We have indicated all parameters in the figures.

3) We thank the referee for pointing this out. At jamming excitations are opening of contacts. In other words when a system of jammed spheres is perturbed (for example strained as was done in Combe and Roux, PRL 85, 3628 (2000)) the dynamics is intermittent and avalanches happen whenever a force crosses zero. This corresponds to having a contact becoming a positive gap and creating a floppy mode. For linear spheres above jamming one can have additional excitations. In particular, in addition to contact becoming positive gaps, one could have contacts becoming overlaps. This corresponds to a force that becomes larger then one). Again, this creates a floppy mode which triggers an avalanche. For the purely linear perceptron case this has been discussed in a recent preprint S. Franz, A. Sclocchi, P. Urbani, arxiv:arXiv:2010.02158. In the present case, with the piecewise linear potential, one can open both contacts in $h=0$ or contacts in $h=-H_0$. These contacts can become either overlaps of all sorts or positive gaps Therefore one can trigger avalanches in many more ways with respect to jamming or the purely linear case. We have included a discussion in the introduction to clarify this point that was previously treated only in the concluding section.

4) We thank the referee, we mentioned that we use the fact that the sign of $\mu$ is directly linked to the convexity of the landscape when discussing isostaticity.

5) We included the number of samples used to produce the statistics.

6) We thank the referee. In the end we found useful to discuss a bit more in depth the smoothed potential since we performed simulations also with the L-BFGS algorithm that runs on that. With this addition in Section 3 (see Eq. (6)) we can now discuss more in detail the form of the Hessian. We hope that this makes the presentation clearer. We also added a reference to our (with S. Franz) previous work on spheres as suggested by the reviewer.

7) For what concerns the exponents in Eq.(11) that appear always the same on all sides of the singularities we note that we have the same as in the purely linear case. In that case, the singularity at $h=0$ has two power laws that are the same on both sides. Here we find the same phenomenology with exactly the same exponents in $h=-H_0$. For what concerns the prefactors: we believe that this numerical symmetry is purely accidental. Indeed we decided to look at the statistics of forces and gaps at a value of the pressure that is such that the number of contacts in $h=0$ is approximately equal to the ones that are at $h=-H_0$. This is because we wanted to have a point in which there is enough statistics to see the behavior of gaps and forces at both corners of the interaction potential. This point is therefore rather symmetric and if one looks at Figure 4 and Figure 6 we can clearly see that. However it is clear that when moving around the phase diagram the density of gaps smaller than $-H_0$ changes and, for example, this decreases to zero when going towards jamming. Therefore we do not expect any universal symmetry for the prefactors. We included in Fig.5 the cumulative distributions of gaps at pressure $p=2$ where the accidental numerical symmetry is lost and the prefactors appear to be different.

8) We thank the reviewer for asking to add the relevant references, we included them in the revisited version of the manuscript.

9) We thank the referee for point out that what we wrote was not clear. The referee is right. One can consider non analyticities that give rise either to discontinuities in the forces or to singularities, for example forces diverging at a point. As an example one could consider models of the following type v(h)=|h|^{1/2}\theta(-h) as well as $v(h)=|\ln|h|| \theta(-h)$. We mentioned that in the conclusions.

10) we apologize for the typos and thank the referee for pointing them out.

11) We did not compute in detail the phase diagram of the model which needs a replica treatment but we expect that it is different from the purely linear case. A simple yet rather clear evidence for this is the following. The topology trivialization transition, for $alpha=4$ happens when $\mu$ changes sign and in the piecewise linear case this happens at $\sigma -\sigma_J\simeq 1.2$. The jamming transition is at $\sigma_J\simeq -0.4$ and therefore the topology trivialization transition is at $\sigma_{dAT} \simeq 0.8$. In the purely linear case the topology trivialization at $alpha=4$ is at $\sigma\simeq 0.6$. Note that the jamming point for the algorithm we are using does not change between the purely linear and the piecewise linear model.

REPORT 2

We thank the reviewer for the critical assessment of our manuscript. The requested changes as well the comments have been very useful to improve the clarity of the manuscript and we are very grateful for that.

Below we give a point by point list of answers to the requested changes.

1) We clarify what is the connection between the power laws and pseudogaps observed in the statistics of local minima and the "proliferation of non-linear excitations, avalanches and crackling noise" we discussed in the text. The main reason why the pseudogaps and power laws in the force and gap distribution are important is that these properties control the response of the system to external perturbations. This has been verified at jamming for hard spheres (see the review by Muller and Wyart, Ann. Cond. Mat. Phys. 2015) as well as in the purely linear perceptron case (see S. Franz, A. Sclocchi, P. Urbani, arxiv:arXiv:2010.02158). Imagine to perform a compression of the system. Isostatic systems stay stable up to the point the force balance condition cannot be verified anymore. This happens when forces exit their stability intervals. In our case this happens when forces exit their stability interval (0,1) and (1,2). This instability is not controlled by the Hessian in local minima (this is why we addressed these excitations as "non-linear", since they are not described by harmonic linear response). When one triggers an excitation, the system looses isostaticity and undergoes an avalanche in which there is a rearrangement of contacts. These avalanches have been shown to be large and power law distributed in the linear perceptron case. We did not investigate them in this work even if we believe that they will show up again for the piecewise linear case because of the same mechanisms described in S. Franz, A. Sclocchi, P. Urbani, arxiv:arXiv:2010.02158.

2) We thank the reviewer for the important remark. Indeed the manuscript was unclear on this point. We improved the presentation in the following way. First we clarify the algorithms we used. We first perform a sort of gradient descent minimization (we use the BFGS library to perform the minimization, the BFGS routine is a particularly efficient version of the conjugate gradient method) on the smoothed cost function. We observe that the minima that we find are always isostatic meaning that the sum of the number of contacts in $h=0$ and in $h=-H_0$ is exactly $N$. The fact that the fluctuations of the total number of contacts are essentially zero is in line with what is observed at jamming (see D. Hexner, P. Urbani, F. Zamponi, PRL 2019) as well as in dense, jammed linear spheres, see S. Franz, A. Sclocchi, P. Urbani, SciPost Phys. 9, 012 (2020). Once we established that minima are isostatic, we give the theoretical argument to show that isostaticity is required for having minima that are marginally stable. Finally we use the compression algorithm developed in S. Franz, A. Sclocchi, P. Urbani, arxiv:arXiv:2010.02158 to produce the plots of the number of contacts and the behavior of the Lagrange multiplier $\mu$ as a function of the distance from jamming. Note that this algorithm finds minima that are isostatic when $\mu<0$ and whose average number of contacts in $h=0$ and $h=-H_0$ are the same of the ones found by the BFGS algorithm.

3) About the weakness related to the lack of the replica treatment of the model: we agree with the referee that having the full replica treatment of the model is important. However we feel that this is not a simple step to do even if we would expect that we can follow the same route proposed in S. Franz, A. Sclocchi, and P. Urbani, PRL 123, 115702 (2019). We first should construct the phase diagram, and then build up the machinery of full-replica-symmetry-breaking (fRSB). Then we should find the fRSB scaling solution along the lines of S. Franz, A. Sclocchi, and P. Urbani, PRL 123, 115702 (2019). The program is clear but not straigthforward and we wanted to focus on the numerical findings. We leave the replica treatment for future work. One of the other reasons why we did not push the replica approach is related to the criticism the referee raises. Indeed our algorithms may find minima that are not close to the ground state (strictly speaking it is unclear where the minima are and a systematic exploration of the landscape is needed in order to make this point clear). However we agree that it may be that the minima we are sampling are far, high-energy, excited states. As we discussed in the discussion section of the manuscript, the replica treatment could be twisted to study excited states since one could show that a family of message passing algorithms is tracked by the replica equations and therefore should land on minima whose properties are the one predicted through replicas, see Ref. [20-22]. However the algorithms we use are not of this type and the referee is right in underlining that the RSB construction would not be enough to reply to the question on why such algorithms seem to find minima with properties described by replicas. We do not have much to say about that, apart that all seems to be as the scaling theory (that encodes the universal features of gaps and forces) that emerges from replicas should describe the landscape that is accessible by local algorithms and therefore should have a strong universal sense. We do not have any proof of this conjecture for the moment. In S. Franz, A. Sclocchi, P. Urbani, arxiv:arXiv:2010.02158 it has been proposed a tentative route to attack this problem.

---

## Editorial Decision

published